

# Alternative dietary protein and water temperature influence the skin and gut microbial communities of yellowtail kingfish (*Seriola lalandi*)

Jack Horlick[1], Mark A. Booth[2] and Sasha G. Tetu[1]

[1] Department of Molecular Science, Faculty of Science and Engineering, Macquarie University, Sydney, NSW, Australia
[2] NSW Department of Primary Industries, Port Stephens Fisheries Institute, Nelson Bay, NSW, Australia

Corresponding author
Sasha G. Tetu, sasha.tetu@mq.edu.au

## ABSTRACT

Fish skin and gut microbiomes contribute to host health and growth and are often significantly different in aquaculture-reared fish compared to wild fish. Determining how factors associated with aquaculture, including altered diet and abiotic conditions, affect the microbiome will assist with optimizing farming practices and non-invasively assessing fish health. Here, juvenile yellowtail kingfish (*Seriola lalandi*) housed at optimal (22 °C) and non-optimal (26 °C) water temperature were fed a fishmeal control diet or the same diet substituted with 30% soy-protein concentrate (SPC) in order to investigate impacts on host health and the microbial community composition of the skin mucosa, gut mucosa and digesta. Each of these sites was observed to have a distinct microbiome composition. The combination of SPC and housing at 26 °C significantly reduced weight gain in yellowtail kingfish and affected immune parameters. The overall microbial composition and relative abundance of specific operational taxonomic units (OTUs) was also significantly altered by inclusion of SPC at 26 °C, with a notable increase in an OTU identified as Photobacterium in the skin mucosa and digesta. Increased relative abundance of *Photobacterium* sp. was significantly correlated with reduced levels of digesta myeloperoxidase in yellowtail kingfish; a recognized innate immunity defense mechanism. The changes in the microbial communities of yellowtail kingfish fed a diet containing 30% SPC at 26 °C highlights the importance of considering the interactive effects of diet and environmental factors on microbiome health in farmed yellowtail kingfish.

## INTRODUCTION

Per capita fish consumption has doubled in the last 50 years and aquaculture now provides over half of all fish for human consumption (77 m tonnes in 2015) (*FAO, 2017*, *2016*). There has been a corresponding increase in the production of aquafeeds and the demand

for fishmeal (FM); the traditional source of protein in aquafeeds (*Cao et al., 2015*; *Klinger & Naylor, 2012*). FM has historically been a key dietary component of the carnivorous farmed fish diet and is generally required in high proportions for optimal health and growth of certain species (*Bendiksen et al., 2011*; *Ringø et al., 2016*). The growth in farming of popular, pelagic carnivores, such as yellowtail kingfish (*Seriola lalandi*), has substantially contributed to global FM consumption (*Kobayashi et al., 2015*). The increasing demand for FM cannot be sustained by wild fishery stocks and therefore diversification of feed sources is an economic and environmental imperative for the aquaculture industry (*Kobayashi et al., 2015*; *Tacon & Metian, 2015*).

Many plant-based products, including those derived from soybean, have emerged as potential substitutes for FM in a variety of carnivorous and omnivorous fish species (*Kissinger, García-Ortega & Trushenski, 2016*; *Hartviksen et al., 2014*; *Zhou et al., 2018*). Methods of raw diet material refinement, such as cooking, grinding or concentration can greatly improve the quality of raw diet materials for fish (e.g., soy protein concentrate (SPC)), leading to better nutritional outcomes and improved fish health (*Bansemer et al., 2015*; *Stone et al., 2017*). However, despite refinement, the dietary inclusion of plant-based feed materials has been linked to gut epithelial damage, reduced nutrient digestibility, and changes in fish immune status which can lead to increased instances of disease such as enteritis (*Bansemer et al., 2015*; *Bowyer et al., 2013*; *Hardy, 2010*). While the cause of these negative outcomes for fish remain unclear, there is evidence that dietary-induced shifts in the fish microbiome are involved (*Bruce, Neiger & Brown, 2018*; *Schmidt et al., 2016*; *Gajardo et al., 2017*; *Legrand et al., 2019*). Indeed, studies in a range of fish species have shown that microbial composition of the gut is significantly influenced by the inclusion of plant-based materials in aquaculture feeds (*Zhou et al., 2018*; *Desai et al., 2012*; *Estruch et al., 2015*; *Green, Smullen & Barnes, 2013*).

Numerous studies have indicated that the microbiome plays an important role in fish health (see detailed reviews (*Legrand et al., 2019*; *Tarnecki et al., 2017*; *Egerton et al., 2018*; *Banerjee & Ray, 2017*)). However, our understanding of how diet-induced shifts in the farmed fish microbiome influence digestion, immunity, and the development of gastrointestinal disorders remains incomplete (*Ringø et al., 2016*; *Legrand et al., 2019*). Whilst progress has been made in certain species, such as Atlantic salmon (*Salmo salar*) and rainbow trout (*Oncorhynchus mykiss*), the effect of diet on the microbiome can vary widely between different fish species (*Ringø et al., 2016*; *Zhou et al., 2018*; *Schmidt et al., 2016*; *Wang et al., 2017*) and little is known for species in the *Seriola* genus.

Yellowtail kingfish are an important commercial species both in Australia and worldwide and minimal information is available regarding the microbiome of this species. In larvae, a recent study has shown that changes in diet result in significant shifts in microbial composition and predicted microbial functionality (*Wilkes Walburn et al., 2019*). In more mature juveniles, reports are limited to two studies of the digesta microbiome (*Ramírez & Romero, 2017*; *Soriano et al., 2018*) and a single study of the gill and skin microbiome (*Legrand et al., 2018*). These studies provide a first overview of the skin and digesta microbiomes of yellowtail kingfish, indicating they are dominated by *Proteobacteria*, as well as varying proportions of *Bacteroidetes*, *Actinobacteria* and

*Firmicutes*, similar to what has been found in other marine fish species (*Tarnecki et al., 2017*; *Ramírez & Romero, 2017*; *Legrand et al., 2018*). These studies also showed that the digesta microbiome differs between yellowtail kingfish of wild and aquaculture origin and that the skin microbiome is impacted by health status, indicating that the rearing environment, such as water temperature, and diet selection are factors likely to influence the microbial composition of different regions of the fish (*Ramírez & Romero, 2017*; *Legrand et al., 2018*). A recent study has also indicated the potential role of specific bacteria in the yellowtail kingfish microbiome in preventing microbial disease (*Ramírez, Rojas & Romero, 2019*). To our knowledge, there are no published studies to date investigating the gut mucosal microbiome of yellowtail kingfish.

In this work we examined the effects of inclusion of FM and SPC on the microbiome of juvenile yellowtail kingfish reared at optimal (22 °C) and non-optimal (26 °C) water temperature. We analyzed how shifts in the microbial composition of skin, gut and digesta of yellowtail kingfish were correlated with measured changes in selected biometric responses and immune-related health parameters. We also provide a detailed overview of the microbial composition of juvenile yellowtail kingfish across multiple body sites, substantially increasing what is known about the microbiome of this commercially important fish species.

## MATERIALS AND METHODS

### Overview of experiment

Triplicate groups of yellowtail kingfish were fed a control diet based on FM or the same diet substituted with 30% SPC. A 30% substitution was chosen as it has been previously shown that soy inclusion above this ratio significantly impacts growth in juvenile yellowtail kingfish, with the driver of reduced growth remaining unknown (*Bowyer et al., 2013*). Each diet was fed to fish at two water temperatures; an optimal temperature of 22 °C and a supra-optimal temperature of 26 °C The optimal and non-optimal rearing temperature were based upon previous work by *Pirozzi & Booth (2009)*. The orthogonal combination of diet type and water temperature resulted in four treatments; hereafter 22 FM and three experimental treatments 22 SPC, 26 FM and 26 SPC. The experiment was done at the NSW Department of Primary Industries (NSW DPI) Port Stephens Fisheries Institute (Taylors Beach, NSW, Australia) in accordance with laws and ethics requirements regarding nutrition research with fish as overseen by the Department of Primary Industries (Fisheries) Animal Care and Ethics Committee (NSW DPI ACEC Authority 93/5).

### Composition of experimental diets

The FM control diet was composed mainly of prime FM (68%) and wheat flour (27%) with a small amount of fish oil (3.5%). The SPC diet was made by blending the FM mash and SPC in a 70:30 ratio; however, supplements such choline chloride, Vitamin-C and vitamin-mineral premix were kept constant between diets. The crude protein and gross energy content of the both FM and SPC diets were approximately 58% and 20.8 MJ kg$^{-1}$, respectively. The formulation, proximate composition and amino acid composition of the diets are detailed in Table S1. All raw diet materials were supplied by Ridley Aquafeed

Pty Ltd. (Robart Court, Narangba, QLD, Australia) and eight mm diameter pellets were manufactured using a twin-screw extruder (MPF24:25; Baker Perkins, Peterborough, UK), oven dried at 45 °C until moisture content was <12%, cooled and stored frozen (−17 °C). Diets were manufactured at the CSIRO Bribie Island Research Centre (Woorim, QLD, Australia).

## Feeding protocols

Mixed sex yellowtail kingfish were obtained from the NSW DPI Marine Fish Hatchery at PSFI and on-grown at low densities in a recirculating aquaculture system. Immediately prior to the experiment fish were being fed a 6 mm commercial aquafeed (Skretting Australia, Cambridge, TAS, Australia). A schematic representation of the experimental flow-through systems used to conduct the feeding trial is detailed in Fig. S1. Briefly, each flow-through system was supplied with raw estuarine water obtained from the Tilligerry Creek adjacent PSFI. This water was firstly filtered through a series of sand filters, a cartridge particle filter and finally two 5 μm bag filters before being transferred to separate supply tanks. Each flow-through system consisted of two 10,000 L supply tanks (storing filtered water) fitted with refrigerated temperature control units and an oxygen diffuser. Temperature controlled water from each supply tank was pumped at constant rate of 6–7 L min$^{-1}$ to six 200 L independent cylindrical polythene tanks. All effluent water was directed to waste.

Seven fish, graded from a larger population, were allocated to each 200 L tank ensuring consistent tank biomass (individual fish weight = 245 ± 18 g and tank biomass = 1,719 ± 32 g). Fish were individually fin-clipped on stocking for subsequent identification. Fish were then gradually acclimated to the temperature regimes over eight days and were handfed a six mm commercial aquafeed (Skretting, Cambridge, TAS, Australia) once daily (11:00 h) at 2% body weight. This ration was slightly restrictive to ensure that all feed was consumed, resulting in equal feed across all treatments. On day nine, fish were lightly anesthetized (10 mg L$^{-1}$, AQUI-S$^{®}$), identified by fin-clip, and individually measured for fork length and weight. Two fish from each tank were selected at this time for baseline health and microbiome data. Each diet was then randomly assigned to triplicate experimental tanks in each temperature system. Fish were handfed test diets once daily (11:00 h) at 3% of the measured tank biomass for the first for 9 days. After 10 days a new feed amount was calculated for each tank based on the expected weight gain of fish and a feed conversion ratio (FCR) of 1.0. This change was based on growth data published on juvenile yellowtail kingfish reared on soy-based diets (*Bowyer et al., 2013*). The same procedure was adopted to increase the quantum of feed required for the remaining 11 days of the trial. The trial was concluded after 21 days at which time all remaining fish were sampled. Rearing water quality parameters were recorded twice daily and are detailed in Table S2.

## Sampling procedures

Microbial communities were sampled from the skin mucosa (skin), distal digesta (digesta), and adhered distal gut mucosa (gut mucosa) of individual animals. Each fish was

transferred to a sterilized stainless-steel benchtop and euthanized by Ikejime. The skin mucus from the right side of the fish between the pectoral and caudal fin was gently scraped with a sterile scalpel blade and the mucus transferred to a sterile 1.5 mL tube and immediately stored on ice. Blood was drawn from the caudal vein at the base of the tail using a 23 G sterile hypodermic needle and transferred to a one mL K3E KEDTA Minicollect tube (Grenier Bio-one, Kremsmünster, Austria) and immediately stored on ice. The fork length and weight of the animal was then recorded before the exterior surfaces were sterilized 100% ethanol. The fish was dissected and the intestine and viscera were carefully removed. The fish, less viscera, and the liver were individually weighed. The distal gut digesta was aseptically excised into a sterile 1.5 mL tube through gently squeezing the gut so as not to include the mucosa and immediately stored on ice. The intestine was opened lengthways and gently washed in sterile phosphate buffered saline (0.01 M) to remove non-adhered luminal gut contents. The gut mucosa was gently scraped using a sterile scalpel blade and the mucosa transferred to a sterile 1.5 mL tube before immediate storage on ice. The microbiota samples were then transferred to and stored in a −80 °C freezer until analyzed. Blood samples were centrifuged at 11,300 rpm for 14 min and the plasma transferred to a sterile 1.5 mL tube and stored at −20 °C until further analysis. A 45 mL aliquot of filtered input estuarine water, in replicate, was sampled on the final day of the trial and stored at −80 °C until further analysis.

## Plasma lysozyme activity

Plasma lysozyme activity was determined by a turbidimetric assay utilizing lyophilized *Micrococcus lysodeikticus* cells (Sigma, Rowville, VIC, Australia) using a method modified from *Cha et al. (2008)*. Lysozyme from chicken egg white (Sigma, Rowville, VIC, Australia) was used as a standard. In a 96 well plate, 15 µL of plasma samples diluted 1:40 in 0.02 M sodium citrate buffer were added to 150 µL of *M. lysodeikticus* suspended in 0.02 M sodium citrate buffer at a concentration of 0.2 mg/mL. The absorbance was immediately measured at 450 nm, and subsequent measurements were taken every 5 min for 60 min. A unit of lysozyme activity was defined as the quantity of plasma required to reduce absorbance by 0.001/min (111).

## Digesta myeloperoxidase activity

The myeloperoxidase (MPO) activity of digesta samples was determined using a colorimetric assay utilizing 3,3′,5,5′-tetramethylbenzidine hydrochloride (TMB) (Sigma, Rowville, VIC, Australia), following the method of Quade and Roth (1995) with minor modifications (*Quade & Roth, 1997*). To prepare equal solutions of digesta, 3 µL of 0.02M sodium citrate buffer was added for each mg of sample (c.250 mg). Samples were heated to 55 °C for 2 min and briefly vortexed and this was repeated twice Samples were centrifuged at 3,000×*g* for 5 min and the supernatant collected for further analysis. Five µL of digesta supernatant were added to 95 µL of Hanks' Balanced Salt Solution without $Ca^{2+}$ and $Mg^{2+}$ (Thermo Fisher Scientific, Scoresby, VIC, Australia) in a 96 well plate. Next, 35 µL of freshly prepared 20 mM TMB and five mM $H_2O_2$ was added to each well and the

reaction stopped after 2 min by addition of 35 µL of 4 M sulfuric acid. The optical density was read at 450 nm.

## DNA extraction and sequencing

DNA extractions were performed using the FastDNA spin kit (MP Biomedicals, Australia) with additional isopropanol purification steps, following the method of *Hart et al. (2015)*. Water samples were filtered through a Sterivex™ GP 0.22 µm filter unit (Millipore, Australia) to isolate the bacteria from the water. The filter was removed and processed in the same manner as the other microbiome samples.

Following extraction, the V4 region of the 16S rRNA gene was amplified using 515 forward and 806 reverse primers with custom barcodes, based upon the earth microbiome primer protocols (*Gilbert, Jansson & Knight, 2014*; *Caporaso et al., 2011*). PCR amplification was performed on 1:50 dilutions of extracted DNA using MyFi Mix (Bioline, Eveleigh, NSW, Australia) with a primer concentration of 400 nM in a final volume of 30 µL. Samples were PCR amplified with 35 cycles at 95 °C for 15 s, 50 °C for 15 s and 72 °C for 60 s. Previously isolated bacterial genomic DNA was used as a positive control and DNAse free water as a negative control. Samples were quantified using Quant-iT™ PicoGreen® (Invitrogen, Scoresby, VIC, Australia). Barcoded amplicons were pooled at equimolar concentrations and gel purified using the Wizard® SV gel and PCR clean up system (Promega, Sydney, NSW, Australia). Pooled amplicons were sequenced using the Illumina MiSeq platform (MiSeq V2 2 × 250 bp paired-end sequencing run) at the Ramaciotti Centre for Genomics, Sydney, Australia.

## PCR assay to screen for plpV and sequencing of PCR products

A region of the plpV gene was amplified using primers described by *Vences et al. (2017)*. PCR amplification was performed on 1:50 dilutions of extracted DNA using MyFi Mix (Bioline, Eveleigh, NSW, Australia) with a primer concentration of 400 nM in a final volume of 30 µL. Samples were PCR amplified with 35 cycles at 95 °C for 30 s, 54.5 °C for 30 s and 72 °C for 60 s. Five microliters of resulting amplicons were visualized on 2% agarose gels (Bioline, Eveleigh, NSW, Australia) using a 1 kbp Hyperladder (Bioline, Eveleigh, NSW, Australia). Three samples were selected to confirm the identity of the amplified product, with Sanger sequencing of bands gel purified using a Wizard® SV gel and PCR clean up system (Promega, Sydney, NSW, Australia), carried out by Macrogen (Seoul, South Korea). Sequence identity was checked by blastn and blastx searches of the NCBI nucleotide and protein databases using Geneious 11.1.5 (https://www.geneious.com) (*Altschul et al., 1990*; *States & Gish, 1994*).

## Bioinformatics and statistical analyses

Raw sequences were demultiplexed by the Ramaciotti Centre for Genomics, Sydney, Australia. Demultiplexed sequences were processed using Quantitative Insights Into Microbial Ecology 2 (QIIME2) software (version 2018.4) (*Caporaso et al., 2010*). Quality control was performed within QIIME2 using DADA2 (*Callahan et al., 2016*). No truncation of the forward or reverse reads was required based upon the quality scores.
Taxonomy was assigned using the QIIME2 q2-feature-classifier plugin and a Naïve Bayes classifier that was trained on the SILVA 99% OTU database trimmed to the V4 region of the 16S rRNA gene (*Caporaso et al., 2010*; *Quast et al., 2013*). Samples with a total number of reads less than 10,000 were discarded from further analysis. Operational taxonomic units (OTUs) from mitochondria and chloroplasts were removed. Rarefaction plots of samples have been included in Fig. S2.

Statistical calculations and graphical construction analysis of amplicon sequence data were performed in the R Studio statistical package (version 1.1.453). Alpha diversity analyses were performed in the phyloseq R package (V.1.24.0) using multifactor analysis of variance (ANOVA) followed by a post hoc Tukey honest significant difference test (*McMurdie & Holmes, 2013*). Non-metric multidimensional scaling (nMDS) plots were constructed using the phyloseq R package (V.1.24.0) (*McMurdie & Holmes, 2013*). The phyloseq R package (V.2.5-2) was used to perform permutational multivariate analysis of variance (PERMANOVA) with 999 permutations on Bray–Curtis distance matrices (V.1.24.0) and to normalize the data used to produce Venn diagrams (*Dixon, 2003*).

Differential abundance of OTUs between treatments were identified using the linear discriminant analysis (LDA) effect size (LEfSe) method, available at http://huttenhower. sph.harvard.edu/galaxy/ (*Afgan et al., 2016*). OTU relative abundance with treatments as the classes of subjects was used as the input. Alpha values of 0.05 were used for the factorial Kruskal–Wallis sum test and the pairwise Wilcoxen test between classes. A threshold of 2.0 was chosen for logarithmic LDA scores.

Spearman correlations, corrected for multiple inference using Holm's method, and node weightings for network analysis were calculated using R package Hmisc (rcorr.adjust) (*Harrell, 2004*). Correlations were considered significant when the correlation $p$-value was $< 0.05$. Significant correlations were visualized in Cytoscape v.3.6.1 (*Shannon et al., 2003*).

Significant differences between treatments for the biometric responses were determined by two-way ANOVA. The fixed factors were diet type (FM vs SPC) and water temperature (22 °C vs 26 °C). Each treatment was applied in triplicate. If ANOVA proved significant ($p < 0.05$) a Tukeys honestly significant difference test was used to separate the treatment means. Statistical analysis was done using GraphPad Prism (version 7) software (GraphPad Software, San Diego, CA, USA).

### Data deposition

The 16S rRNA gene sequence data generated and analyzed in this study can be found in the GenBank Sequence Read Archive (SRA) database under accession number PRJNA492935.

## RESULTS

### Analyses of biometric responses and immune parameters

Triplicate groups of Yellowtail kingfish were fed FM and SPC based diets at optimal (22 °C) and non-optimal (26 °C) temperatures. The orthogonal combination of diet type and water temperature resulted in four treatments; hereafter a control treatment

**Table 1 Growth performance, feed conversion ratio, Fulton's body condition, and hepatosomatic index of yellowtail kingfish for each treatment.**

| Factors[e] | 22 FM | 22 SPC | 26 FM | 26 SPC |
|---|---|---|---|---|
| Weight gain (%)[a] | 70.37 ± 2.8% | 62.68 ± 3.21% | **59.26 ± 5.14%** | **54.04 ± 4.87%** |
| Weight gain (g)[a] | 201 g ± 26 g | 181 g ± 41 g | **173 g ± 54 g** | **155 g ± 45 g** |
| SGR (% BW day$^{-1}$)[b] | 1.76 ± 0.21 | 1.64 ± 0.20 | 1.52 ± 0.42 | 1.41 ± 0.41 |
| Fork length growth (%)[a] | 17.38 ± 0.76% | 16.72 ± 1.57% | 13.35 ± 0.97% | 13.07 ± 1.54% |
| Feed conversion ratio[c] | 0.93 ± 0.02 | 0.99 ± 0.02 | 1.09 ± 0.06 | **1.19 ± 0.04** |
| Fulton's body condition[d] | 1.43 ± 0.03 | 1.42 ± 0.01 | 1.45 ± 0.01 | 1.49 ± 0.02 |
| Hepatosomatic index[e] | 1.07 ± 0.04 | 1.03 ± 0.03 | 1.18 ± 0.04 | 1.16 ± 0.03 |

Notes:
[a] Means of triplicate tanks (five fish per tank) ± standard deviation.
[b] Specific growth rate (% body weight per day) = (ln (final weight) − ln (initial weight)) × 100/days.
[c] Feed conversion ratio = feed given (g)/weight gain (g).
[d] Fulton's body condition = 100 × body weight (g)/length (cm)$^3$.
[e] Hepatosomatic index = 100 × liver weight (g)/body weight (g).

(22 FM) and three experimental treatments (22 SPC, 26 FM and 26 SPC). The survival rate of fish across all treatments was 100% and no obvious signs of illness were noted on internal and external examination of sampled specimens. Specific growth rate (weight) of fish, shown in Table 1, was found to be significantly affected by both water temperature ($p = 0.0017$ and $F = 21.24$) and by diet ($p = 0.049$ and $F = 5.36$); however, there was no interaction between the water temperature and diet factors ($p = 0.87$ and $F = 0.03$) when analyzed by two way ANOVA. Relative weight gain was significantly lower in fish reared at 26 °C, irrespective of the diet fish were fed (Table 1). An increase (worsening) in feed conversion ratio (FCR) of was observed for all experimental treatments compared to 22 FM, but was only significant for 26 SPC (Table 1). Feed conversion ratio was calculated as feed given (g) divided by weight gain (g). No adjustment was made for waste as all feed was consumed. Fulton's condition factor and the hepatosomatic index of fish were not significantly affected by water temperature and diet or their interaction (Table 1).

Assays were performed to measure components of the innate immune system in digesta and plasma samples. Digesta myeloperoxidase (MPO), which is produced by neutrophils, was significantly lower (one way ANOVA; $p = 0.12$ and $F = 1.055$) in fish fed SPC at 22 °C and 26 °C compared to fish fed the 22 FM control treatment (Fig. 1A). Plasma lysozyme levels were significantly higher in fish fed SPC at 26 °C compared to fish fed the FM diet at 22 °C (Fig. 1B). However, the 26 SPC treatment was not significantly different to the groups fed SPC at 22 °C or FM at 26 °C (Fig. 1B).

## Microbiome sequencing and assignment of OTUs

A total of 84 juvenile yellowtail kingfish were sampled resulting in 48 gut mucosal samples, 59 digesta samples and 55 skin mucosal samples following quality filtering (Table 2). DNA extractions were performed on an additional 36 gut mucosal samples, 25 digesta samples and 29 skin mucosal samples; however, these did not yield visible PCR products or failed to yield reads upon sequencing. Fish body site samples had on average ≥47,556 reads, while water samples had an average of 33,371 reads following sequence quality

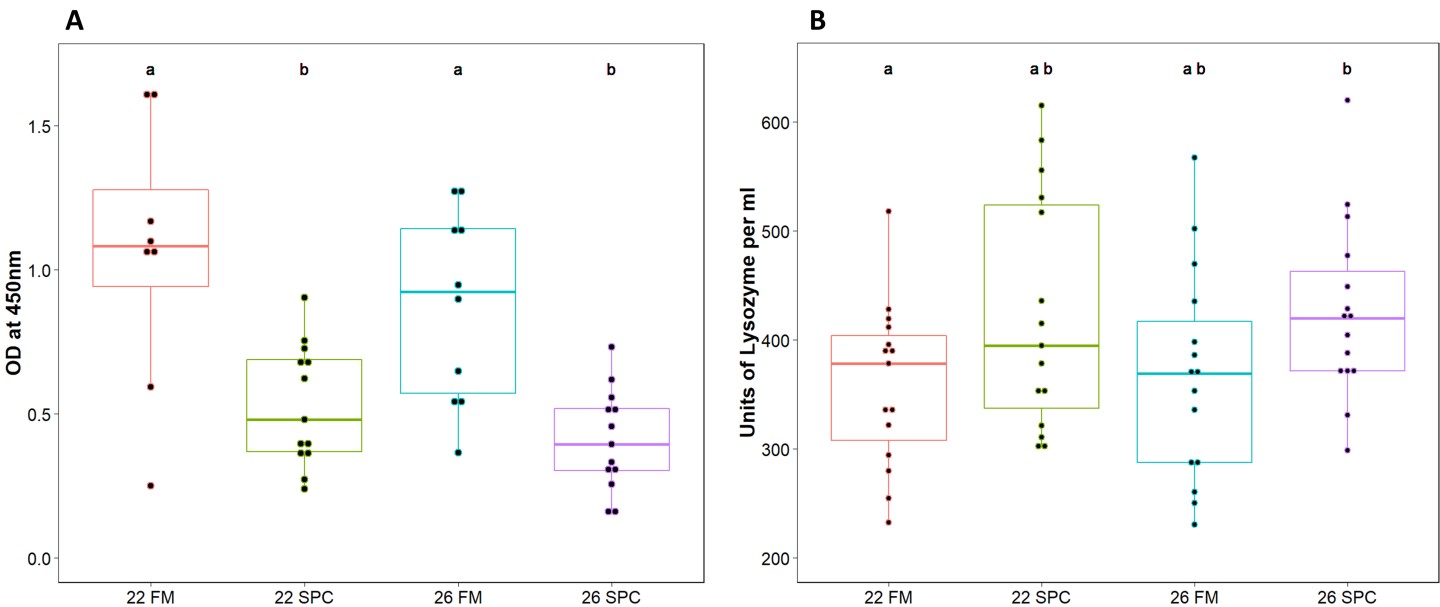

**Figure 1 Levels of (A) digesta MPO and (B) plasma lysozyme in yellowtail kingfish for each treatment.** Boxplot with points representing individual samples and the box representing the median and interquartile range. Different letters denote statistically significant ($p < 0.05$) differences between treatments by one way ANOVA.

**Table 2 Sample numbers, sequencing reads, and OTU statistics post quality control filtering for fish body sites and filtered estuarine water[a].**

| Sampling site | Sampling time point | Baseline samples | Total reads | Average reads per sample | Total OTUs | Unique OTUs |
|---|---|---|---|---|---|---|
| Gut mucosa | Baseline | 15 | 886,485 | 59,099 ± 24,628 | 388 | |
| | Final | 33 | 2,097,015 | 63,545 ± 32,000 | 318 | |
| | Total | 48 | 2,983,500 | 62,156 ± 29,702 | 531 | 84 |
| Digesta | Baseline | 16 | 485,621 | 44,147 ± 4,482 | 162 | |
| | Final | 43 | 2,320,220 | 48,337 ± 11,925 | 355 | |
| | Total | 59 | 2,805,841 | 47,556 ± 11,019 | 418 | 112 |
| Skin | Baseline | 15 | 706,277 | 47,085 ± 15,650 | 973 | |
| | Final | 41 | 2,025,615 | 49,405 ± 20,197 | 1,232 | |
| | Total | 56 | 2,731,892 | 48,784 ± 18,976 | 1,476 | 985 |
| Rearing Water | Final | 2 | 66,741 | 33,371 ± 4,060 | 97 | 5 |
| Total | | 165 | 8,587,974 | 52,048 ± 21,486 | 1,712 | |

Note:
[a] Number of samples, total reads, mean reads (± SD), total OTUs, and unique OTUs for the gut mucosa, digesta, skin and water samples.

control and filtering (Table 2). A total of 7,787 unique sequence variants were detected, which were assigned to 1,713 OTUs' at a 99% similarity level.

Pairwise permutation multivariate analysis of variance tests (PERMANOVA) confirmed that there were no significant differences between the baseline microbial community profiles of different tanks of fish at the beginning of the experiment (Table S3). PERMANOVA showed no significant influence of tank on the microbial composition

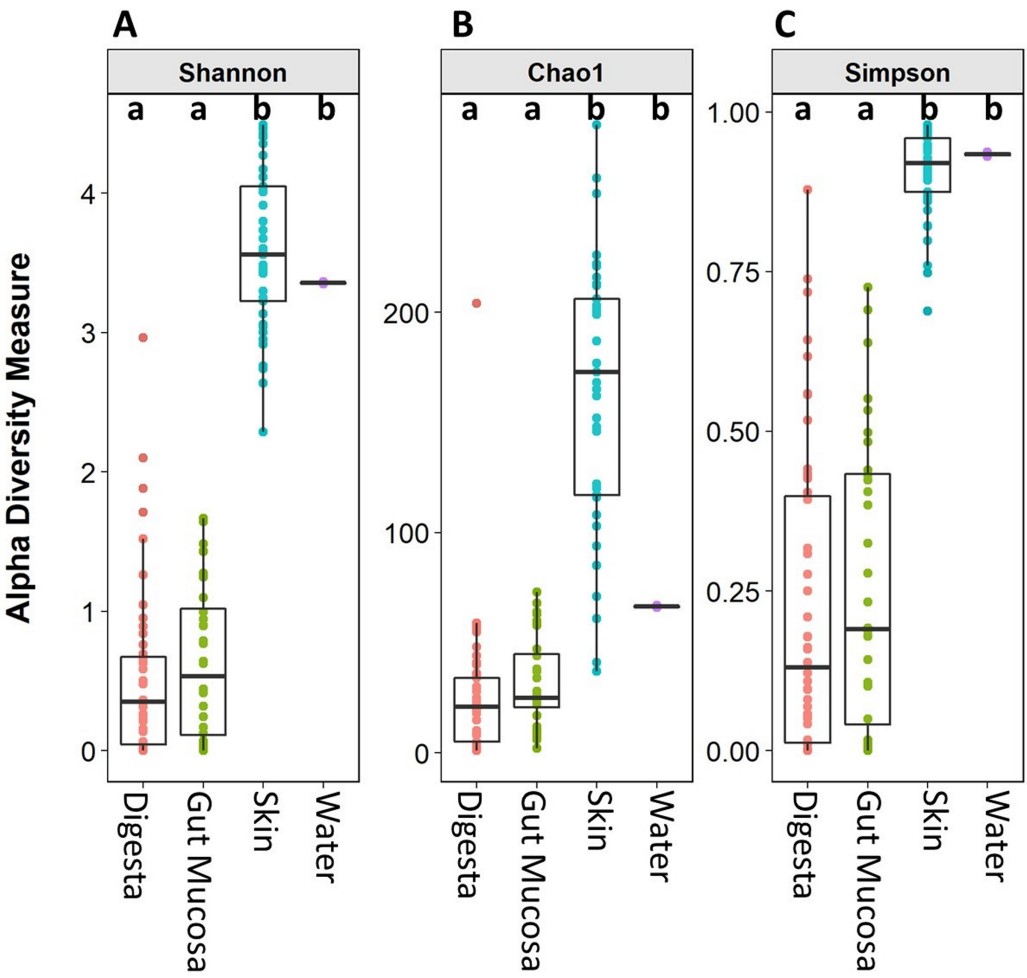

**Figure 2 Alpha diversity (Shannon), richness (Chao1), and evenness (Simpson) of digesta, gut mucosa, skin, and water microbial communities.** Boxplots of (A) Chao1, (B) Shannon and (C) Simpsons alpha diversity metrics for each sample, grouped by sampling site. Points represent individual samples and the box represents the median and interquartile range Different letters denote significant ($p < 0.05$) differences between treatments by ANOVA.

within each treatment for any of the body sites, indicating that tank was not a significant factor driving community changes (Table S4).

## Comparison of microbial composition across body sites

Microbiome alpha diversity measures were calculated for each of the body sites and the water. The microbial community richness (Chao1) and diversity (Simpson and Shannon) observed in the skin samples were significantly higher than that of the gut mucosa and digesta. There was no significant difference in the alpha diversity measures between the digesta and gut mucosa (Fig. 2). The alpha diversity measures in the rearing water were more similar to the skin than the internal gut microbiome samples (Fig. 2).

Of the 1,713 OTUs identified, the rearing water shared 288 OTUs with the skin, while there were 274 OTUs in common between water and gut mucosa and 239 shared with the

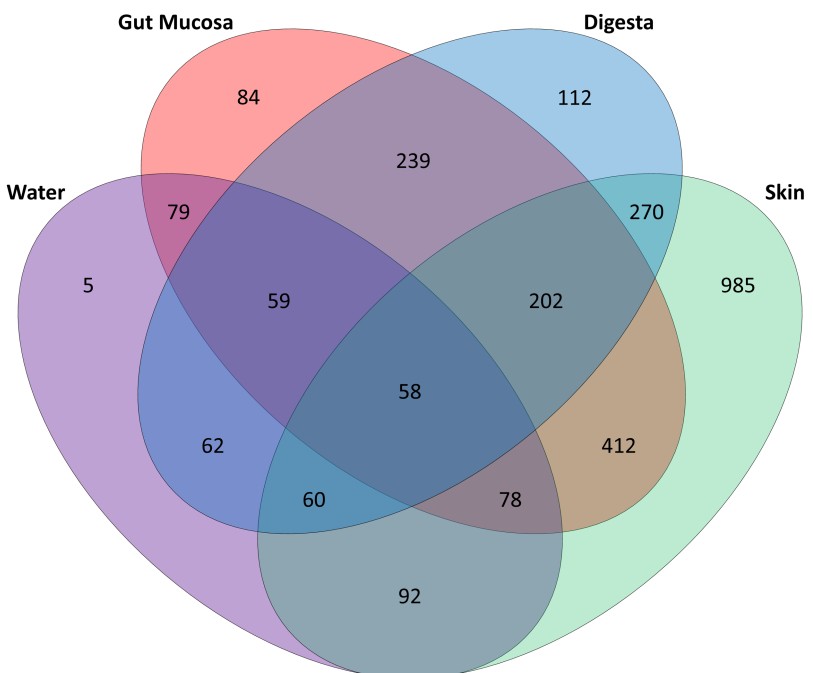

**Figure 3 Distribution of OTUs identified in the gut mucosa, digesta, skin and water samples.** Venn diagram detailing the number of OTUs that are unique to and shared between each of body sites and water.

**Table 3 Pairwise PERMANOVA comparing the microbial communities of gut mucosa, digesta, and skin samples by treatment[a].**

| Body site | Body site | 22 FM | | 22 SPC | | 26 FM | | 26 SPC | |
|---|---|---|---|---|---|---|---|---|---|
| | | $R^2$ | $p$ | $R^2$ | $p$ | $R^2$ | $p$ | $R^2$ | $p$ |
| Gut Mucosa | Digesta | 0.083 | 0.125 | 0.124 | **0.025** | 0.053 | 0.428 | 0.217 | **0.002** |
| Gut Mucosa | Skin | 0.285 | **0.001** | 0.231 | **0.001** | 0.181 | **0.002** | 0.138 | **0.002** |
| Digesta | Skin | 0.255 | **0.001** | 0.408 | **0.001** | 0.252 | **0.002** | 0.435 | **0.001** |

**Note:**
[a] Pairwise PERMANOVA with 999 permutations was performed on a Bray–Curtis dissimilarity matrix. Comparisons for which $p < 0.05$ are presented in bold.

digesta and only 5 OTUs which were specific to the rearing water (Fig. 3). The skin contained the highest number of unique OTUs (985) with digesta and gut mucosa sites containing only 112 and 84 unique OTUs respectively. The skin and gut mucosa housed 750 common OTUs, while the digesta microbiome shared somewhat lower numbers of OTUs with the other sites (590 OTUS shared with skin and 558 OTUs in common with gut mucosa microbiomes). Of the core microbiome observed across all body sites (260 OTUs), 53% were assigned as *Proteobacteria*, 15% as *Firmicutes*, 14% as *Actinobacteria*, 8% as *Bacteroidetes*, 2% as *Cyanobacteria*, with the remaining 8% split between 16 additional phyla.

Microbial community composition was observed to be strongly influenced by body site (PERMANOVA analysis, Table 3). The community composition of the skin was

**Table 4 Pairwise PERMANOVA comparing the microbial communities of experimental treatments to the control in the skin[a].**

| Control | Treatment | Differ by | $R^2$ | $p$ |
|---------|-----------|-----------|-------|-----|
| 22 FM | 22 SPC | Diet | 0.107 | **0.005** |
| 22 FM | 26 FM | Temp | 0.083 | **0.008** |
| 22 FM | 26 SPC | Diet + Temp | 0.117 | **0.003** |

**Note:**
[a] Pairwise PERMANOVA with 999 permutations was performed on a Bray–Curtis dissimilarity matrix. Comparisons for which $p < 0.05$ are presented in bold.

significantly different from that of the gut mucosa and digesta in all four treatments (Table 3). Based on PERMANOVA, the gut mucosa and digesta bacterial communities were not significantly different in fish fed FM diets. However, in fish fed SPC diets, the gut mucosa and digesta microbial communities were significantly different (Table 3).

## Skin microbiome response to altered diet and water temperature

The overall community microbial composition of the skin samples was affected by both diet and water temperature. We observed significant differences in the skin microbiomes of fish fed FM and SPC diets, based on PERMANOVA analysis (Table 4). Elevated water temperature also affected the microbial communities, with a significant difference between the 22 FM control treatment and the 26 FM treatments (Table 4). The combination of elevated water temperature and inclusion of SPC had the greatest impact on the skin microbiome, as this treatment was associated with the strongest shift in community composition compared to the 22 FM control treatment (Table 4).

The skin microbiome in all treatments was dominated by four phyla, which together contributed a combined relative abundance of >90%. These phyla, in order of relative abundance, were *Proteobacteria*, *Bacteroidetes*, *Firmicutes* and *Actinobacteria* (Fig. S3). At the genus level only four genera contributed >5% of relative abundance of the skin microbiome: *Ralstonia*, *Photobacterium*, unassigned bacterium of the family *Rhodobacteraceae* and *Thalassotalea* (Fig. 4). Approximately 35% of the inferred microbial population in each treatment was assigned to these four genera. The remaining population was comprised of 949 lower abundance genera, of which 916 had a relative abundance of less than 1%.

To investigate which OTUs significantly contributed to the differences in skin microbiome composition associated with diet and temperature, differential abundance testing with linear discriminant analysis (LDA), effect size (LEfse) analysis was performed. We found that 25, 23 and 12 OTUs were significantly differentially abundant in the 22 SPC, 26 FM and 26 SPC treatments respectively when compared to the control treatment (Fig. 5). For all three treatments, the relative abundance of OTUs assigned to *Photobacterium* and *Lawsonella* were increased when compared to the control treatment, while OTUs assigned to *Chitinophagales* and *Salegentibacter* were signficantly decreased. The 22 SPC treatment was associated with increased abundance of several OTUs that were not significantly increased in the other treatments, including *Acinetobacter*, *Streptococcus*, *Haemophilus* and *Micrococcus* representatives (Fig. 5).

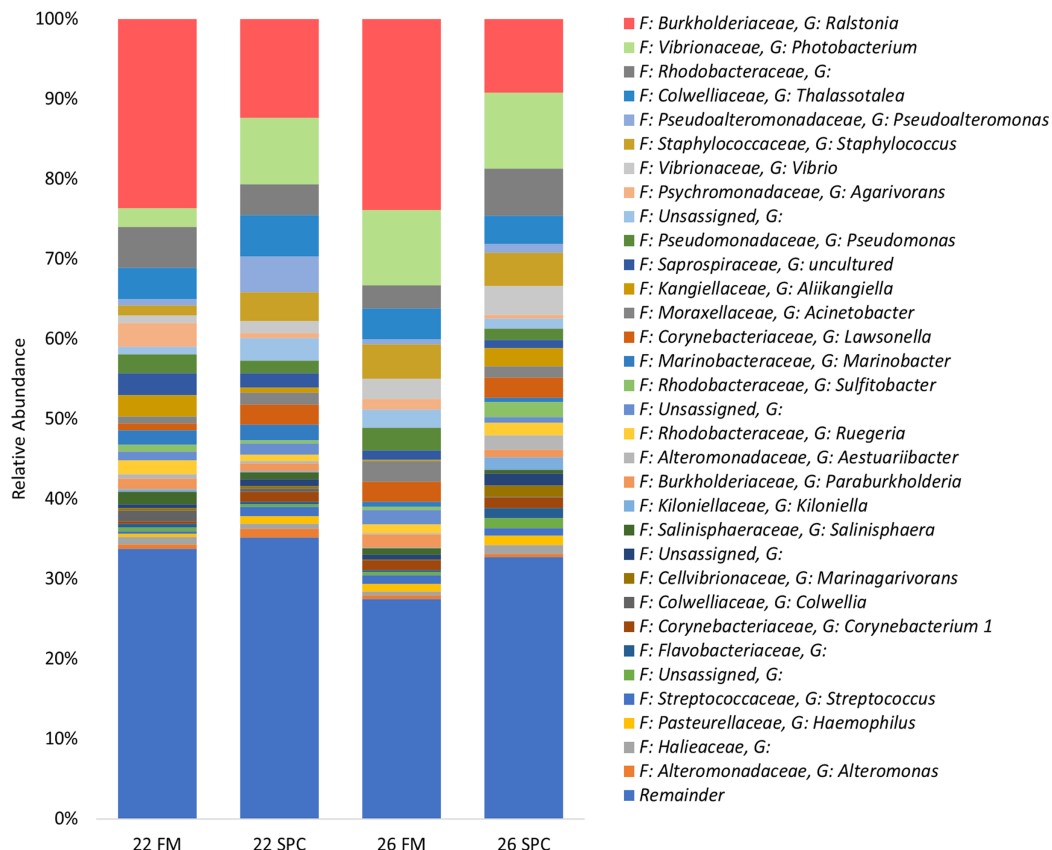

**Figure 4 Skin microbial composition (mean relative abundance of OTUs) for each treatment grouped by genus with assigned family detailed.** Where no genus has been taxonomically assigned, the genus is left blank. Families that comprise less than 1% of the population are grouped together under remainder.

The 26 SPC treatment was associated with increased levels of only two OTUs, assigned as *Lawsonella* and *Photobacterium*, whereas the 22 SPC and 26 FM treatments were associated with higher levels of 21 and 6 OTUs respectively when compared to the control treatment. While these OTUs were found to be significantly differentially abundant between treatments, considerable variability was still observed in the response of individuals within the treatments (Fig. 5). However, the variability of individuals within each treatment did not appear to be impacted by which tank fish were housed (Fig. 5).

## Digesta microbiome response to altered diet and water temperature

As with the skin, the digesta microbial communities of fish fed SPC at 22 °C were significantly different from the 22 FM control when analyzed by PERMANOVA (Table 5). Similar differences were noted in fish housed at 26 °C, which had significantly altered community profiles compared to the 22 FM control treatment (Table 5). As with the skin, the most pronounced shift in digesta community composition from the control treatment was noted in fish fed SPC at 26 °C (Table 5).

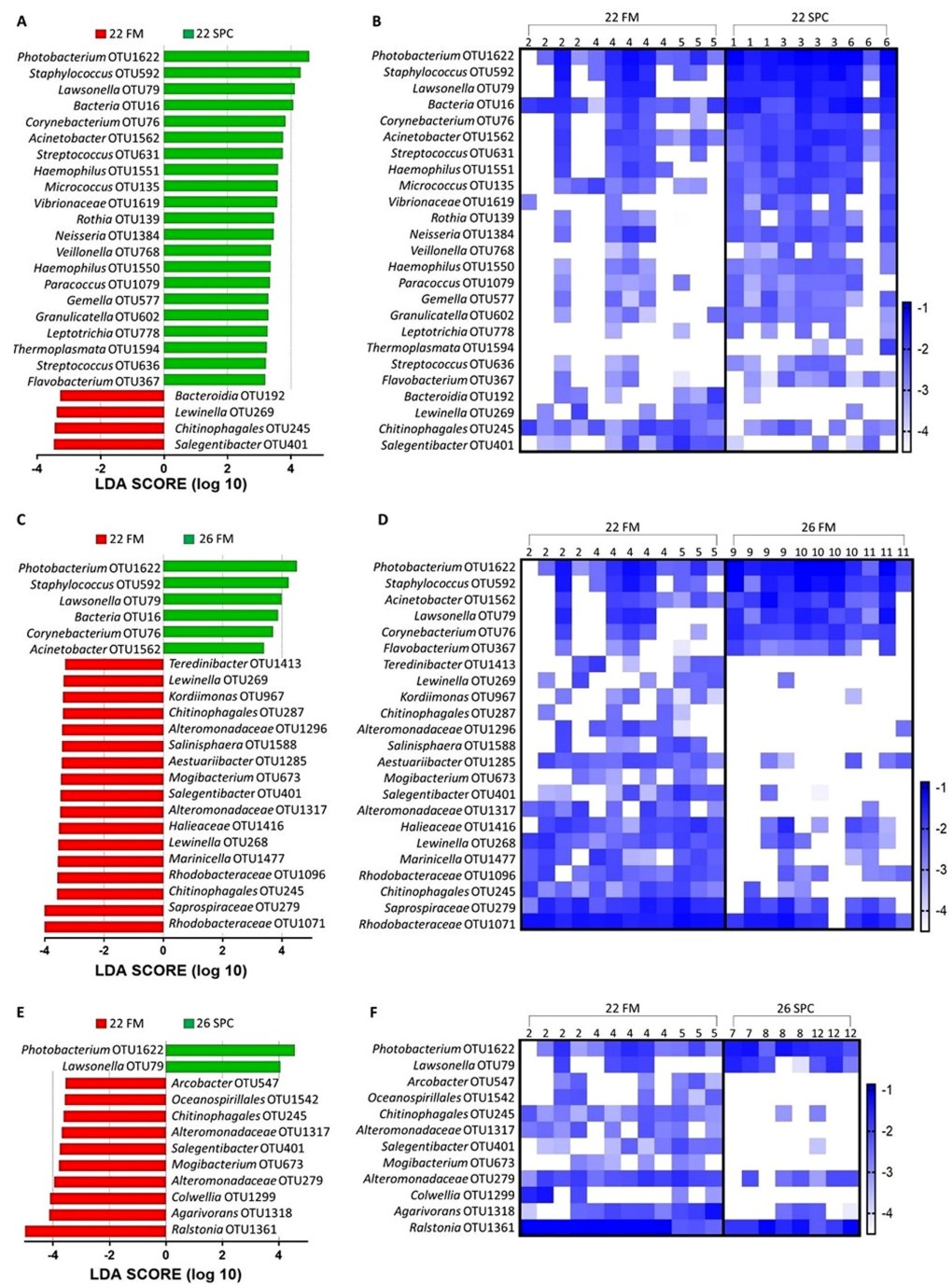

**Figure 5 Impact of diet and temperature on the abundance of OTUs in the skin microbiome.** Differentially abundant OTUs were determined by LEfse analysis between control treatment and 22 SPC, 26 FM and 26 SPC. (A), (C) and (D) show the significant ($p < 0.05$) LDA scores calculated for each OTU. (B), (D) and (F) show the relative abundance ($Log_{10}$ transformed) for each biological sample. Rows of the heat maps correspond to OTUs and columns to biological samples. Biological samples in the heat map are grouped by treatment and are labeled with their corresponding tank number. Blue and white denote the highest and lowest relative abundance respectively. OTUs were assigned at a genus level where possible, otherwise, the lowest inferred taxonomic level available was given.

**Table 5 Pairwise PERMANOVA comparing the microbial communities of experimental treatments to the control in the digesta[a].**

| Control | Treatment | Differ by | $R^2$ | $p$ |
|---|---|---|---|---|
| 22 FM | 22 SPC | Diet | 0.173 | **0.005** |
| 22 FM | 26 FM | Temp | 0.118 | **0.029** |
| 22 FM | 26 SPC | Diet + Temp | 0.267 | **0.001** |

**Note:**
[a] Pairwise PERMANOVA with 999 permutations was performed on a Bray–Curtis dissimilarity matrix. Comparisons for which $p < 0.05$ are presented in bold.

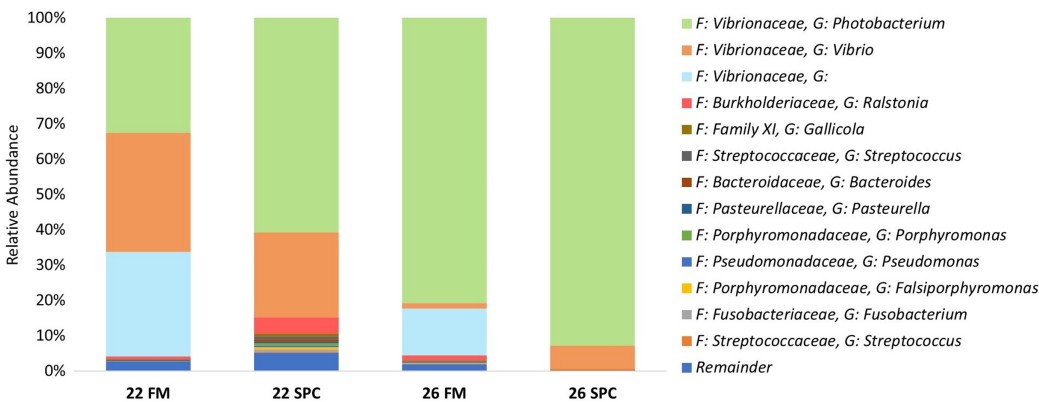

**Figure 6 Digesta microbial composition (mean relative abundance of OTUs) for each treatment grouped by genus level with assigned family detailed.** Where no genus has been taxonomically assigned, the genus is left blank. Families that comprise less than 0.5% of the population in all treatments are grouped together under remainder.

Digesta microbial communities were dominated by the family *Vibrionaceae*, with a relative abundance of over 80% across all four treatments (Fig. 6). No other family comprised more than 5% of the population in any treatment. Whilst the relative abundance of the family *Vibrionaceae* was relatively consistent across treatments, specific genera within this family were strongly impacted by both diet and temperature, with an OTU assigned to the genus *Photobacterium* dominating the communities across the experimental treatments relative to the control treatment (Fig. 6). To establish how the digesta microbiome changes in respect to altered diet and elevated temperature, the set of OTUs showing significant differences between treatments was determined. LEfse analysis revealed 5, 3 and 10 differentially abundant OTUs in the 22 SPC, 26 FM and 26 SPC treatments respectively when compared to the control treatment (Fig. 7). Across each of the three experimental treatments, the relative abundance of OTU 1,622, assigned to genus *Photobacterium*, was significantly increased with respect to the control treatment (Fig. 7). The combination of 26 °C and SPC diet was associated with significantly lower abundance of nine OTUs (relative to the control treatment), whereas the 22 SPC and 26 FM treatments were both associated with a decrease in one OTU (Fig. 7). Within each treatment there was no common trend in the relative abundance of OTUs for each tank, indicating that the housing tank was not a major driver of the differences in microbial composition between samples (Fig. 7).

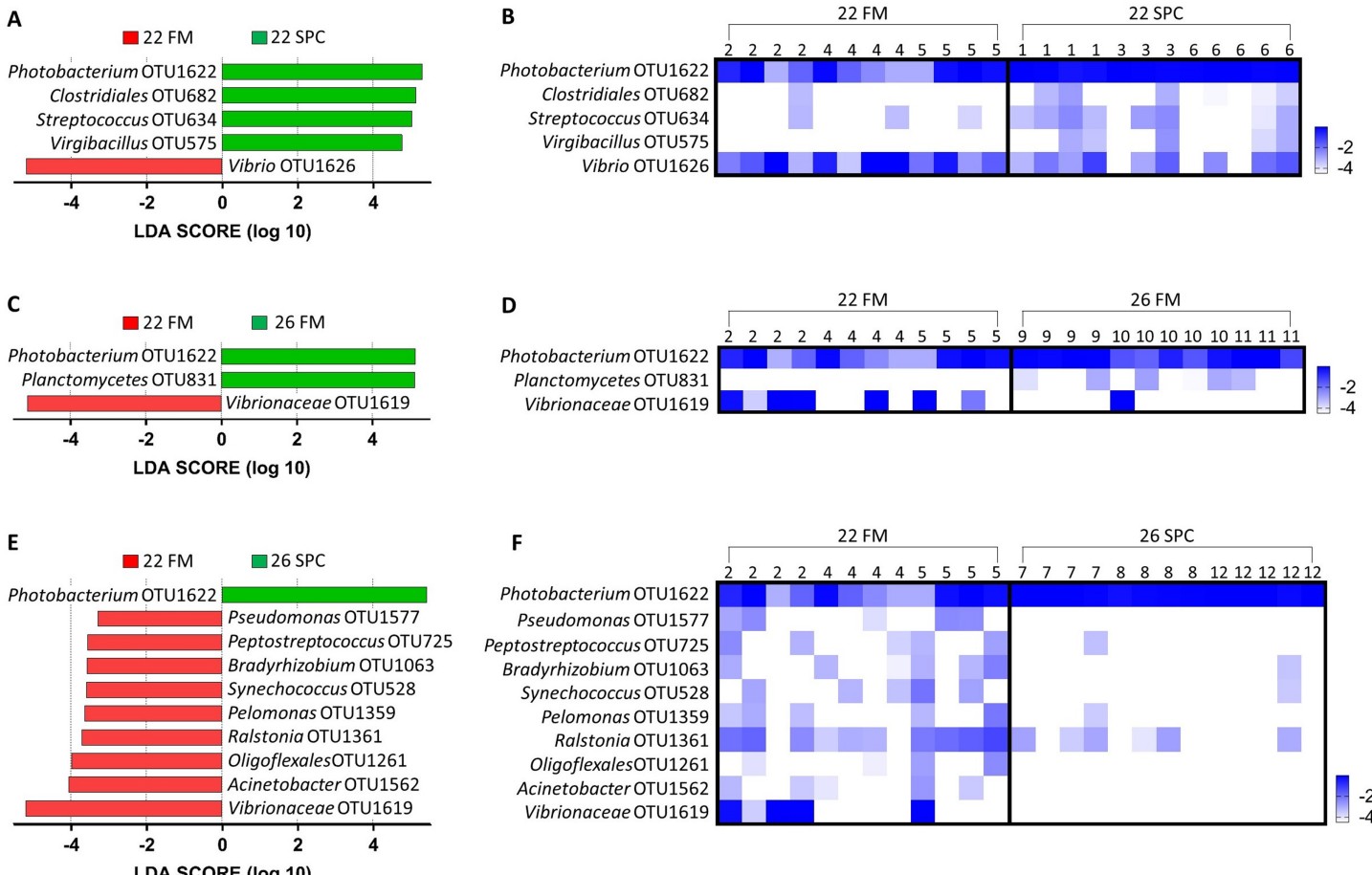

**Figure 7 Impact of diet and temperature on the abundance of OTUs in the digesta microbiome.** Differentially abundant OTUs were determined by LEfse analysis between control treatment and 22 SPC, 26 FM and 26 SPC. (A), (C) and (D) show the significant ($p < 0.05$) LDA scores calculated for each OTU. (B), (D) and (F) show the relative abundance (Log$_{10}$ transformed) for each biological sample. Rows of the heat maps correspond to OTUs and columns to biological samples. Biological samples in the heat map are grouped by treatment and are labeled with their corresponding tank number. Blue and white denote the highest and lowest relative abundance respectively. OTUs were assigned at a genus level where possible, otherwise, the lowest inferred taxonomic level available was given.

Operational taxonomic unit 1,622, assigned to genus *Photobacterium*, was dominated (94%) by a single sequence variant. A blastN search of the NCBI nucleotide database (July 2018) showed this sequence variant shares 100% identity only with characterized strains of species *Photobacterium damselae*, including *Photobacterium damselae* subsp. *damselae* (GenBank ID MG077071.1) and *Photobacterium damselae* subsp. *piscicida* (GenBank ID MH472944.1) which have been reported as fish pathogens (*Do Vale, Marques & Silva, 2003*; *Xie et al., 2007*). Screening for the *P. damselae* hemolytic phospholipase gene *plpV*, a marker of virulent strains of this species (1), was carried out using the 43 endpoint digesta samples. All 38 digesta samples that had a relative abundance of OTU 1,622 greater than 20% were PCR positive for this marker, while no PCR product was observed in the five samples with a relative abundance of OTU 1,622 less than 2%. Sequence confirmation performed on three positive samples showed that all shared >96% nucleotide identity and 100% translated amino acid identity with PlpV in

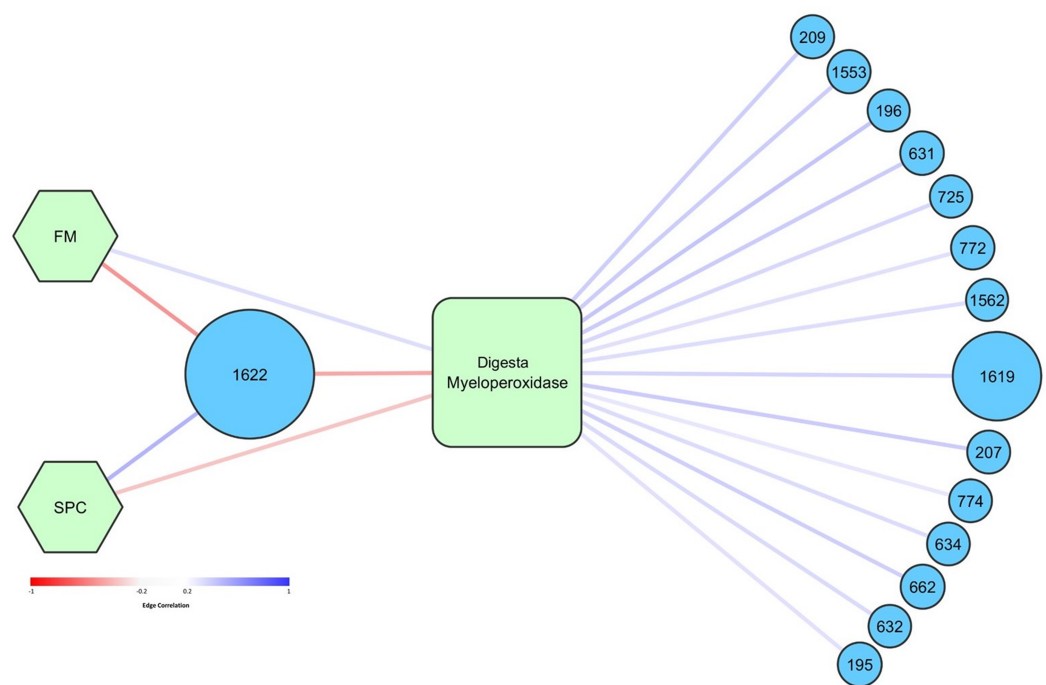

**Figure 8 Graphical network representing the interactions in digesta samples between OTUs, diet, and digesta myeloperoxidase.** Only significant correlations ($p < 0.05$) are shown. Circular nodes correspond to OTUs and the size of these nodes represents the maximum relative abundance of that OTU across all digesta samples. Square nodes correspond to physiological measurements taken, and hexagonal nodes to diets. Red edges represent negative correlations, whilst blue edges represent positive correlations. The strength of the red or blue color represents the strength of the correlation. Taxonomy assigned to OTUs is presented in Table S4.               

*Photobacterium damselae* subsp. *piscicida* and *Photobacterium damselae* subsp. *piscicida* (Tables S5 and S6). A blastN search of the NCBI nucleotide database (July 2018) of the representative sequences of OTUs 1,619 and 1,626 (both also assigned to the family *Vibrionaceae*) did not provide a reliable indication of taxonomy at a more detailed level as both sequences matched a number of different *Vibrio* species.

Network analysis was performed to determine how specific OTUs correlated with diet, temperature, and measured health and growth parameters. Higher abundance of OTU 1622, tentatively thought to represent pathogenic *Photobacterium damselae*, was significantly correlated with lower levels of digesta MPO (Fig. 8). OTU 1,622 was also significantly positively correlated with the SPC diet (and negatively with FM diet) (Fig. 8). OTU 1,622 was the only OTU that was significantly correlated with diet ($p < 0.05$). There was a positive association between the FM diet and higher levels of digesta MPO. Fourteen OTUs were positively correlated with increased levels of digesta MPO (Fig. 8). The most abundant of these was OTU 1,619, classified to family *Vibrionaceae*, which had a maximum relative abundance of 66% across all digesta samples. The other 13 OTUs all had a relative abundance of less than 4% in any given sample and were diverse in their taxonomic classification, spanning seven different taxonomic orders (Table S7).

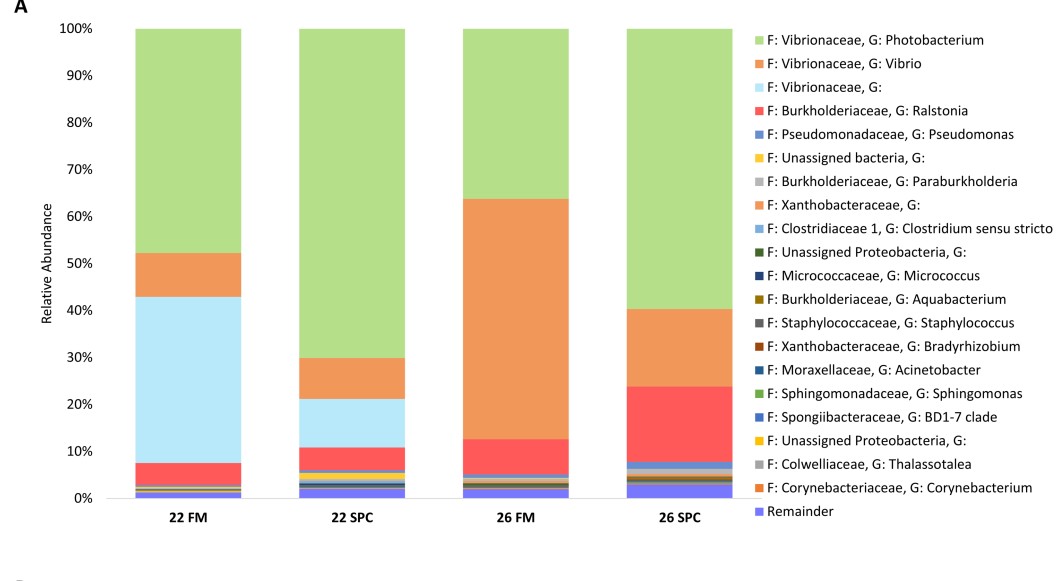

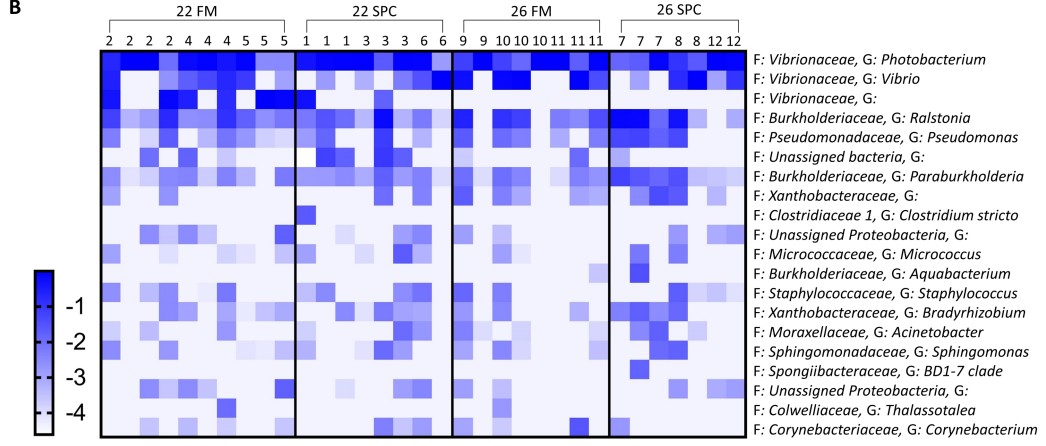

**Figure 9 Mean relative abundance of OTUs present in gut mucosal samples grouped by genus.** (A) Mean relative abundance of OTUs in each treatment grouped by genera with family detailed. Genera comprising less than 0.2% grouped under remainder. (B) Mean relative abundance ($Log_{10}$ transformed) of OTUs grouped by genera that comprise >0.2% of each treatment. Rows of the heat map correspond to OTUs and columns to biological samples. Biological samples in the heat map are grouped by treatment and are labeled with their corresponding tank number. Blue and white denote the highest and lowest relative abundance respectively. Where no genus has been taxonomically assigned, the genus is left blank.

## Gut mucosa microbiome response to altered diet and water temperature

Compared to the skin and digesta communities, the gut mucosa microbiome was found to be less affected by alterations in diet and temperature. The overall gut mucosa microbial community showed no significant differences between the treatments when analyzed by PERMANOVA (Table S8). In all treatments, the gut mucosa microbiome was dominated by three genera of the family *Vibrionaceae* (Fig. 9A). These three genera had a combined relative abundance of 91% in 22 FM, 89% in 22 SPC, 87% in 26 FM and 76% in 26 SPC treatments (Fig. 9A). The only other genus that comprised over 2% of the

population in any of the treatments was *Ralstonia*, with the remainder of the population assigned to 264 genera with an abundance of less than 2% (Fig. 9A). While there were clear differences in the mean relative abundance of different genera by treatment, individuals within each treatment showed a high degree of variability in genera level relative abundance (Fig. 9B). LEfse analysis revealed only one OTU that had a significantly different relative abundance between treatments. OTU 1,619, which is only classified to the family level of Vibrionaceae was statistically significantly lower in the 22 SPC (LDA −5.5), 26 FM (LDA −5.4), and 26 SPC (LDA −5.2) treatments.

## DISCUSSION

### Diet and temperature impact growth and innate immunity

The dietary inclusion of 30% SPC had a significant impact on the growth of juvenile yellowtail kingfish. This effect was amplified at an elevated water temperature of 26 °C. Similar impacts on weight gain were noted in another study feeding SPC to juvenile yellowtail kingfish (*Bowyer et al., 2013*). Previous studies have shown that the reduced digestibility of soy-based diets can impact the growth and feed conversion ratio of farmed fish (*Deng et al., 2006*). Whilst digestibility was not measured in the current experiment, the feed conversion ratio of the FM and SPC diets at optimal temperature (22 °C) were comparable, indicating that digestibility was likely not a key factor. Irrespective of water temperature, the inclusion of SPC was also associated with changes in immune parameters. Digesta myeloperoxidase (MPO), a component of the fish innate immune system produced by neutrophils (*Palić et al., 2005*), was found to be at significantly lower levels in fish fed SPC when compared to fish fed a FM based diet. In contrast, plasma lysozyme levels were increased in fish fed the SPC diet. Plasma lysozyme is an immune defense molecule that causes lysis of bacteria through breaking linkages in the cell wall (*Saurabh & Sahoo, 2008*). Increased levels of lysozyme activity have also been observed in Atlantic salmon (*Salmo salar*) fed diets containing SPC (*Krogdahl et al., 2000*).

### Microbial composition and diversity of sampled yellowtail kingfish body sites

The overall microbial community composition of the skin mucosa was distinct from the other sampled sites and exhibited significantly higher alpha diversity. Prior work in rainbow trout (*Oncorhynchus mykiss*) and Atlantic salmon has also shown the skin microbiome to be distinct from that of the gut (*Lowrey et al., 2015*; *Gajardo et al., 2016*; *Minniti et al., 2017*) and higher diversity in the skin compared to gut is consistent with previous work in rainbow trout (*Lowrey et al., 2015*). The skin alpha diversity metrics seen in our study are also consistent with previous findings for yellowtail kingfish (*Legrand et al., 2018*); however, the alpha diversity of the digesta microbiota was not as low as reported here. It is noteworthy that a recent study investigating the gut microbiome of wild Atlantic cod (*Gadus morhua*) also found similarly low levels of gut diversity and reported that the microbiome was dominated by an OTU assigned as *Photobacterium* (*Riiser et al., 2018*). It is possible, therefore, that the genus *Photobacterium* is able to rapidly colonize and become dominant in the gut of certain fish species.

*Proteobacteria* were predominant among the set of OTUs common to all sites followed by varying proportions of *Bacteroidetes*, *Firmicutes* and *Actinobacteria*. This is broadly consistent with previous yellowtail kingfish microbiome research where these four phyla were found to be highly abundant in both the skin and digesta microbiomes (*Ramírez & Romero, 2017*; *Legrand et al., 2018*). Work in other farmed fish species has also found *Proteobacteria* to be the dominant phylum in the microbiome, followed by differing proportions of *Firmicutes, Actinobacteria* and *Bacteroidetes* (*Minniti et al., 2017*; *Chiarello et al., 2015*; *Larsen et al., 2013*; *Uren Webster et al., 2018*). At genus level gut mucosal and digesta communities were dominated by OTUs assigned to *Vibrio* and *Photobacterium*. These two genera are reported to be dominant members of carnivorous fish microbiomes across numerous studies, as reviewed recently by *Egerton et al. (2018)*, although it is worth noting 16S rRNA gene copy is relatively high for these genera, which could contribute to some extent to their high relative abundance. In contrast, *Soriano et al. (2018)* did not note high proportions of these genera in yellowtail kingfish reared in saltwater recirculation systems at 22 °C or at 26 °C. This discrepancy highlights the substantial difficulty in obtaining consistent microbial populations across experimental systems, even within the same species and applying similar community analysis methodology.

## The skin microbiome is affected by changes in diet and water temperature

The skin mucosa of yellowtail kingfish was associated with 982 unique OTUs and shared only 92 OTUs with the estuarine water samples, indicating that the skin microbiome is independent of the surrounding water (Fig. 4). This finding is consistent with that of *Chiarello et al. (2015)* who noted that the skin microbial communities of both gilthead seabream (*Sparus aurata*) and European seabass (*Dicentrarchus labrax*) shared few OTUs with the water.

The skin microbiome community composition was significantly altered by both water temperature and diet treatments. Temperature variation is well known to affect bacterial growth rates and particular species or strains may have differences in their optimal growth temperature, however specific temperature preferences are only known for some culturable isolates. The relative abundance of OTUs assigned to *Photobacterium* and *Lawsonella* were increased in the skin mucosa of fish fed SPC at both 22 °C and 26 °C when compared to the control treatment. *Lawsonella* belongs to the family *Corynebacteriaceae* and increases in the abundance of this family have also been previously reported in response to altered diet (*Rimoldi et al., 2018*). *Rimoldi et al. (2018)* noted that the relative abundance of *Corynebacteriaceae* was increased in the gut of rainbow trout fed diets containing high proportions of animal by-product compared to FM diets, indicating that alternative feeds may specifically impact this family. Low levels of *Photobacterium* have been noted in past microbiome surveys of the skin of marine fish; however, this genus showed higher relative abundance in this study than previously reported (*Legrand et al., 2018*; *Minniti et al., 2017*; *Chiarello et al., 2015*; *Siriyappagouder et al., 2018*). The relative abundance of *Photobacterium* was particularly high in the 26 FM,

22 SPC and 26 SPC treatments, indicating that raw diet material selection and temperature can influence the level of these bacteria on the skin. These alterations may be as a result of diet and tememprature induced alterations in the composition of mucins in the skin mucosa which have previously been shown to influence the microbiome and are a key component of this mucosal barrier (*Legrand et al., 2019*; *Merrifield & Rodiles, 2015*; *Tapia-Paniagua et al., 2018*). Diet and temperature related changes in the microbiome may also be due to altered composition of mucosal immunoglobulins, which potentially play a key role in determining which bacterial species inhabit the mucosa (*Cecelia & Irene, 2017*; *Gomez, Sunyer & Salinas, 2013*; *Xu et al., 2013*).

## The skin microbiome as a potential non-invasive biomarker of fish health

Elevated temperature, regardless of diet, was associated with a decrease in the relative abundance of OTUs assigned to *Alteromonadaceae* when compared to the control. Lower relative abundance of *Alteromonadaceae* has previously been associated with poor health, with lower levels observed in yellowtail kingfish exhibiting enteritis and in Atlantic salmon infected with Salmonid alphavirus (*Legrand et al., 2018*; *Reid et al., 2017*). It should also be noted that *Soriano et al. (2018)*, whom made no note of reduced health in yellowtail kingfish fish reared at 26 °C, did not observe altered levels of *Alteromonadaceae*. Given that fish housed at 26 °C in the present study were associated with lower growth and altered immune parameters, the relative abundance of *Alteromonadaceae* in the skin might be a potential indicator of poor fish performance and reduced health status; however, further work would be needed to determine this. Increased prevalence of the genus *Photobacterium* in the skin community might also be used as an indicator of health, possibly in conjunction with *Alteromonadaceae* to non-invasively assess the health status of fish.

## Diet and temperature influence the microbiome of the digesta

The inclusion of SPC at 22 °C resulted in a substantial increase in the relative abundance of OTU 1,622 (assigned to *Photobacterium)* in the digesta relative to the 22 FM control along with increases in three other bacterial OTUs assigned to *Clostridiales* and *Streptococcus*. The combination of SPC at 26 °C was also associated with an increase in OTU 1,622, while nine OTUs were found to be significantly reduced in their relative abundance compared to the control.

Analysis of the sequence variants within OTU 1,622 indicates that this likely represents the species *P. damselae*, but does not distinguish between different strains of this species which are reported to differ in their pathogenicity (*Vences et al., 2017*). Detection of the *plpV* virulence gene in all digesta samples for which OTU 1,622 was abundant (>20% of relative abundance) indicates that this OTU is potentially a representative of pathogenic *P. dameselae*. This suggests that dietary inclusion of SPC, particularly in combination with elevated water temperature may create opportunities for pathogenic strains of *P. damselae* to dominate in the digesta, out-competing other microbial species. It should be noted that there are multiple genes that contribute to pathogenicity in *P. damselae*

(*Altschul et al., 1990*) and therefore the presence of *plpV* is not conclusive evidence for pathogenicity, especially given there were no obvious sign of photobacteriosis in the fish at the end of this trial. Longer-term experiments involving culturing and characterization of predominant digesta bacteria would be useful for further investigating the connection between diet, *P. damselae* abundance and yellowtail kingfish health.

Network analysis showed that high relative abundance of OTU 1,622 was correlated with reduced MPO in the digesta, indicating that this OTU may be influencing MPO production in the gut. Virulent strains of *P. damselae* ssp. *piscicida* have been shown to induce apoptosis of sea bass (*Dicentrarchus labrax* L.) neutrophils, and therefore it is possible that the reduction in measurable MPO is due to *Photobacterium* induced apoptosis of neutrophils (*Do Vale, Marques & Silva, 2003*). However, we were not able to determine whether the observed alterations in the levels of MPO are mediated by changes in the microbiome or are due to a direct impact of the diets we tested. Soy-based diets have previously been shown to induce alterations in the histology of the gut, and therefore may be directly influencing neutrophil recruitment and thus the level of MPO (*Bansemer et al., 2015*; *Bakke-McKellep et al., 2007*). It is also possible that a combination of increased relative abundance of OTU 1,622 and the direct impact of SPC on the gut both acted to influence the levels of MPO in the digesta. The correlation between reduced levels of MPO and significantly higher relative abundance of OTU 1,622 indicates that suppression of the fish immune system through these intimated mechanisms, or indeed other mechanisms, may have contributed to the dominance of the genus *Photobacterium* observed in the digesta. Further studies considering a wider range of immune parameters are required to follow up on these findings.

## The gut mucosal microbiome is less responsive to changes in diet and temperature

Unlike the digesta microbiome, the composition of the gut mucosa microbiome was not strongly affected by the SPC diet or altered water temperature (Table S8). For gut mucosal samples the relative abundance of OTU 1,622 (assigned to *Photobacterium*) did not reach the same levels as in the digesta and no significantly differentially abundant OTUs were observed for any of the treatments. The microbiota of the digesta are considered free-living and transient, in contrast to the gut mucosal community, which has undergone long term co-evolution with the host and is influenced to a greater degree by host physiology (*Banerjee & Ray, 2017*). Indeed, the chemical composition of the gut mucosal layer plays a key role in bacterial adhesion and therefore their ability to colonize the mucosa (*Banerjee & Ray, 2017*). It is possible that the gut mucosa microbiota may be slower to respond to the particular perturbations used in this work and a longer term study may show additional changes in this community.

## CONCLUSIONS

Many previous studies investigating fish microbiomes have involved surveying populations of wild fish and farmed fish (*Tarnecki et al., 2017*; *Ramírez & Romero, 2017*;

*Legrand et al., 2018*). Such studies provide an overview of the microbial communities and can provide insight into how different factors contribute to shaping the microbiome, with environment and diet often being implicated as drivers of microbial composition. However, given there are many differences between the lifestyles of wild and farmed yellowtail kingfish it is has not been possible to determine the degree to which specific factors are responsible for influencing or controlling the composition of the microbiome. Here, we performed a controlled study which measured the separate and combined effects of diet and water temperature on the microbiome of yellowtail kingfish. We observed distinct changes in microbial community profiles as well as significant alterations in fish growth and specific immune parameters. Given that the microbiome in fish has been shown to influence digestion, nutrient assimilation, and stimulation of the immune system in other studies, we suggest that high levels of SPC (30%) and elevated water temperature (26 °C) resulted in shifts in the microbial communities of juvenile yellowtail kingfish which subsequently influenced their growth trajectory and immune status (*Cecelia & Irene, 2017*; *Ray, Ghosh & Ringø, 2012*).

In the digesta of fish fed the SPC diet at 26 °C, a single OTU tentatively identified as pathogenic *P. damselae*, dominated the microbial community. This OTU was correlated with a reduction in levels of the innate immune defense molecule MPO. Given the significantly lower growth observed in this group, as well as the alterations to their digesta microbiome, we recommend limiting the amount of SPC in commercial diets for yellowtail kingfish until further inclusion studies are undertaken.

An elevated water temperature of 26 °C independently influenced the response of the yellowtail kingfish microbiome to changes in diet and also further contributed to lower growth rates. These results indicate that the impact of stressors commonly associated with commercial fish production (such as non-optimal water temperature) should be carefully considered by aquaculture practitioners before they commence widespread use of alternative dietary materials or they make significant changes to the diet of their fish. Many factors such as water temperature are not normally controllable in a commercial fish farm, therefore the seasonal timing of dietary changes will likely be important. Future studies of the impact of alternative raw diet materials would benefit from considering both optimal and non-optimal environmental conditions encountered in commercial aquaculture settings. This might include examination of changing abiotic factors such as salinity, pH and dissolved oxygen.

## ACKNOWLEDGEMENTS

We thank the all the technicians at the Port Stephens Fisheries Institute, including Justin Tierney, Ian Russel, Steve Gamble and Basseer Codabaccus, for all their assistance with the feeding trial. We would also like to thank the NSW DPI for access to the facilities and resources of the Port Stephens Fisheries Institute. We thank Ridley Aquafeed (Michael Salaini) for providing raw materials used to make the diets. We thank Dr. Hasinika Gamage, Dr. Liam Elbourne, and Prof. Ian Paulsen for their assistance and technical support with the bioinformatic analysis.

### Funding

This work was supported by the Macquarie University Master of Research program and the New South Wales Department of Primary Industries. Dr Sasha G. Tetu is supported by an Australian Research Council 'Discovery Early Career Research Award' fellowship (#DE150100009). The funders had no role in study design, data collection and analysis, decision to publish, or preparation of the manuscript.

### Grant Disclosures

The following grant information was disclosed by the authors:
Macquarie University Master of Research Program.
New South Wales Department of Primary Industries.
Australian Research Council 'Discovery Early Career Research Award': #DE150100009.

### Competing Interests

The authors declare that they have no competing interests.

### Author Contributions

- Jack Horlick conceived and designed the experiments, performed the experiments, analyzed the data, prepared figures and/or tables, authored or reviewed drafts of the paper, and approved the final draft.
- Mark A. Booth conceived and designed the experiments, analyzed the data, authored or reviewed drafts of the paper, and approved the final draft.
- Sasha G. Tetu conceived and designed the experiments, analyzed the data, prepared figures and/or tables, authored or reviewed drafts of the paper, and approved the final draft.

### Animal Ethics

The following information was supplied relating to ethical approvals (i.e., approving body and any reference numbers):

All experimental protocols and procedures involving animals were carried out in accordance with the requirements of the Department of Primary Industries (Fisheries) Animal Care and Ethics Committee (NSW DPI ACEC REF 93/5).

### Data Availability

Data is available at the NCBI Short Read Archive: PRJNA492935.

### Supplemental Information

Supplemental information for this article can be found online at http://dx.doi.org/10.7717/peerj.8705#supplemental-information.

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
