# Peer review of "Alternative dietary protein and water temperature influence the skin and gut microbial communities of yellowtail kingfish (Seriola lalandi)"

_PeerJ, doi:10.7717/peerj.8705_

## Round 0.1 · original submission · Major Revisions

The reviewers have commented on your above paper. They indicated that it is not acceptable for publication in its present form.

However, if you feel that you can suitably address the reviewers' comments (included), I invite you to revise and resubmit your manuscript.

·

Basic reporting

The manuscript is well written in a professional language, though I have noticed a few spelling mistakes:

1) Line 101: The authors should specify the control treatment, as stated afterward in line 266 “one control treatment (22 FM) and three experimental treatments (22 SPC, 26 FM and 26 SPC)”. Here is a suggestion: The orthogonal combination of diet type and water temperature resulted in four treatments; hereafter a control treatment (22 FM) and three experimental treatments (22 SPC, 26 FM and 26 SPC).

2)Line 247: spelling mistake “calculated”

3) Line 270: spelling mistake “temperature”

The literature references are appropriate, and the introduction provides sufficient information to understand the aims of the experiment. However, here are of few comments:

1) Line 73: While the authors mention that reports on yellowtail kingfish microbiome are limited to 3 studies, more recent reports have been conducted such as one investigating the impact of diet on the larvae associated microbiota (Wilkes et al 2019, https://doi.org/10.1111/1751-7915.13323). In addition, a report on the selection of native probiotics for yellowtail kingfish was recently published and highlights the potential role of the microbiota in yellowtail kingfish health (Ramirez et al 2019, https://doi.org/10.1007/S12602-019-09550-9).

2) The authors cite 3 review papers to highlight the role of the microbiome in fish health (line 63). However, some of these reviews are mainly focused on structure and diversity rather than on the role of the fish microbiome. In this regard, more appropriate reviews could be cited (e.g. Banarjee and Ray 2017. Bacterial symbiosis in the fish gut and its role in health and metabolism. Symbiosis. https://doi.org/10.1007/s13199-016-0441-8; Legrand et al 2019. A microbial sea of possibilities: current knowledge and prospects for an improved understanding of the fish microbiome. Reviews in Aquaculture. https://doi.org/10.1111/raq.12375).

Experimental design

The aims of the experiment are well defined. The authors examined juvenile yellowtail kingfish microbiota at different body sites (skin, gut mucus and digesta) in response to changes in temperature and diet. They also investigated changes in immune-related health parameters with the aim to correlate them with shifts in the fish microbiota.

Overall, the methods used in this study are robust and well described. Though a relatively large proportion of the samples were not successfully sequenced, the authors had sufficient replicates to perform robust data analysis. The collection of a water sample at the beginning of the experiment would have been valuable to compare the microbial diversity between sample types (Figures 2 and 3). However, it does not impact the conclusions drawn from this work.

Validity of the findings

All important data and information are included in the manuscript. However, a few corrections should be made to improve the quality of the paper:

1) Even though the authors mentioned that only samples with a minimum of 10,000 sequences were kept for downstream analysis (line 231), a rarefaction plot would be useful to show sufficient sequencing depth and should be added in the supplemental files.

2) Significant differences are presented in line 269 and 281 with p-values of 0.12, 0.22 and 0.12 using one-way ANOVA. Since an alpha of 0.05 is commonly used to differentiate significance, authors should provide a table summarising the complete results of the different statistical tests performed (e.g. one-way ANOVA) in order to prove significant differences. If significant differences are not found, authors should change the wording accordingly (e.g. was found to be substantially affected though not significant).

3) The authors generated Spearman correlations using the R package Hmisc (line 247). Significant correlations were then imported in Cytoscape to generate a network. However, to the best of my knowledge, the correlation p-values generated by Hmisc are not corrected and therefore, a multiple testing correction, MTC, (e.g. Bonferroni) is needed before importing significant correlations in Cytoscape (for an example, see Sylvain et al. 2019, https://doi.org/10.1111/mec.15184 ). Since the MTC is likely to influence the number of significant correlations, this would impact the number of nodes and edges of the network which would have to be reconstructed. If MTC was applied and not mentioned by the authors, this should be stated in the methods section.

4) The authors investigated microbial diversity using different indexes. Among them, Simpson was used to evaluate the microbial community evenness (line 304). However, the Simpson index (symbol D) generated by Phyloseq does not reflect the microbial community evenness but instead provide information about diversity and richness (Simpson, E.H. (1949). “Measurement of diversity". Nature. (163): 688). As a result, another index should be used in order to investigate the microbial community evenness such as the Simpson evenness measure (symbol E) or Pielou’s evenness measure.

Additional comments

Horlick et al investigated the effect of soy-protein concentrate and water temperature on juvenile Yellowtail Kingfish microbiota. They found that both water temperature and diet perturbed the skin and gut bacterial communities, associated with changes in immune parameters in some cases. The data generated in this study provides additional information on the yellowtail kingfish microbiota and highlights the need to consider the combined effect of diet and environmental factors on yellowtail kingfish health. Therefore, I believe that this manuscript deserves to be published if the previously cited revisions are made.

Reviewer 2 ·

Basic reporting

The paper by Horlick et al. is an important contribution to the skin and gut microbiome of a cultivated fish species with very limited knowledge on this issue. The paper is based on a very well designed experimental set up and the methods used are some of the most appropriate to date. Finally the paper is adequately structured and its presentation is very reader-driendly despite the ample volume of data/information it contains.

Experimental design

The authors have based their analysis on some of the most appropriate methodological tools available to date. Also, the bioinformatics and statistical analysis they used in just about right, without providing any unnecessary analyses.

Validity of the findings

This is a very important contribution for the field of fish microbiomes. The authors explain all the presented results in a very comprehensive way, supported by the relevant literature and scientifically sound speculations.

Additional comments

I was not able to find any "mistakes" that need to be corrected. For this I have several suggestions but only because the data you analysed are so meaningful, and this is your paper I will leave it as it is. What needs to be included, though, is (a) a short justification of how you chose 26oC and not higher/lower than optimal. and (b) how did you end up using a 30% replacement in the diet (and not 50% or 20% or other)?.
A final comment is about Photobacterium and its increased abundance at 26oC, despite it is known to do better at lower temperatures. It is very hard to distinguish whether its increased abundance is due to the increased temperature or the diet substitution. Another issue relted to this genus is its high number of 16S rRNA copies (as it is for several other Proteobacteria) which ranges between 9 and 21 (mean 15±3.8). If one corrects for such high numbers, then its relative abundance drops significantly. Kormas (2011) suggested that when Proteobacteria dominate in a sample, the effect of multiple 16S rRNA gene copy numbers might impact considerably the big picture of bacterial diversity (Kormas KA (2011) Interpreting diversity of Proteobacteria based on 16S rRNA gene copy number. In: Sezenna ML (ed) Proteobacteria: Phylogeny, metabolic diversity and ecological effects. Nova Publishers, Hauppauge, NY, p. 73-89. ISBN: 978-1-61761-810-9).

Reviewer 3 ·

Basic reporting

The article “Alternative dietary protein and water temperature influence the skin and gut microbial communities of yellowtail kingfish (Seriola lalandi)” contributes with new knowledge regarding the microbiome of yellowtail kingfish. It is well written and clear. However, there are some issues that need to be addressed, specially with the performance results and the microbiome`s statistical analyses.

Experimental design

The experimental design is clear. Nevertheless, there are some point to consider/clarify regarding the experimental design:
- Is there any reason to use only 7 fish per tank? This amount of fish does not represent what may happen in the aquaculture industry at any scale
- Growth performance is only represented by weight gain. This is insufficient to have a clear view of the performance, some other performance parameter should be included in the analyses (SGR, TGR, others). Also, the weight gain was not stated (grams gained), only a difference in % between group. In order to understand results, the weight gain and other performance parameters should be stated (with variation) in the manuscripts and the tables (not only the %)
- It is unclear how the FCR was calculated. Did you collect feed waste? If so, Feed intake should be corrected by feed waste. If feed waste was not collected, this should be stated in the manuscript as a limitation in the results/discussion
- Line 160-161 need clarification of the method of collection of digesta. Could this method affect the results?
- Microbiota analysis should include tanks as a factor in the statistical analysis. If not, you would need to pool the number of samples per tank., as variation among tanks in each treatment should be taking into account. Another alternative is to use a linear mixed model approach using tank as random factor and diet and temperature as fixed factors
- How was the microbiota data filtered regarding contamination during DNA extraction and PCR steps? Was the PCR product of controls sequenced and analyzed? The results from the controls should be considered in order to remove contamination (you can check this article: Simple statistical identification and removal of contaminant sequences in marker-gene and metagenomics data. doi: 10.1186/s40168-018-0605-2).

Validity of the findings

The findings are interesting, although they should modify the points suggested for the experimental design.
Other comments regarding results and findings:
- Line 269-270, weight gain should be specified (with variation)
- Line 269-270 Not significantly affected? Or is there a typo error?
- Line 314 make sure that the OTUs from Cyanobacteria are not coming from chloroplast. Be sure to remove the reads belonging to OTUs assign to chloroplast
- 358-359 unclear. 22FM is the control treatment
- Line 446-455 This is a repetition of the results with no discussion
- Line 524-527 More studies are needed to be able to state this
- Line 560 increase relative abundance, not total abundance
- Line 561-565 Only MPO was measured, it could only be an indication of immune compromise. More studies and several other analyses are needed to validate this finding

Additional comments

- Figures need to have bacterial genera in cursive
- Figure 5 difficult to see, it needs to be turn around
- Figure 8 needs to be bigger
- Line 42-44 need update information, as for now, feed for carnivorous fish such as Atlantic salmon are not dependent on fishmeal

Reviewer 4 ·

Basic reporting

No comment

Experimental design

No comment

Validity of the findings

No comment

Additional comments

The manuscript entitled “Alternative dietary protein and water temperature influence the skin and gut microbial communities of yellowtail kingfishn (Seriola lalandi)” submitted by Horlick et al. reports the effect of fishmeal substitution with 30% soy-protein concentrate (SPC) combined with rearing water temperature on skin and gut microbiome of yellowtail. The present study examines in detail the interactive effects of diet and environmental factors on microbiome health and, partly, on innate immunity of host. In the last years, intestinal microbiome gathered interest of aquaculture scientist since it potentially regulates host health status and thus productivity of aquaculture industries. In fact, in these years and increasing numbers of studies have indicated that the microbiome plays an important role in fish health and diet is considered one of the main factors putting selective pressure on the gastrointestinal microbial composition in vertebrates. Hence this paper deal with a key topic and provides valuable outcomes that should be out in public. In addition, commendable effort that the sequencing data have been deposited on public GenBank Sequence Read Archive (SRA) database. For all these reasons, I recommend this paper for publication, although some relevant changes are necessary to the currently submitted version of the manuscript. Authors should consider the following points carefully and revise the paper.
Major revisions
• The SPC diet formulation reported in Supplementary Table 6 does not correspond with what reported at line 109-110 “The SPC diet was made by blending the FM mash and SPC in a 70:30 ratio”. Indeed from Suppl. Table 6 the SPC diet contains besides FM also wheat flour. Therefore, the ratio 70:30 has to be referred to (FM + Wheat flour):SPC.

• At line 269 authors state that “Relative weight gain of fish was found to be significantly
affected”, however in the text they report a p-value > 0.05. Is it a typing error?
Moreover, concerning weight gain data, they report that no interaction was found between two main factors (diet and temperature), but in table 1 they indicate a statistical significance between 26FM vs 22FM and 26SPC vs 22FM. It has no sense when no interaction diet x temperature has been found.

• Differences between communities (beta diversity) based on Bray Curtis distance should be represented also by PCoA plots in order to better understand and graphically verify the results of PERMANOVA analysis made.

• Since the water microbiota has been analysed, why was not the same done for feed-associate microbial communities? It is known that undigested feed could influence transient gut microbial community composition

Minor points
Figures 1, 5, 6-9 are difficult to read I suggest increasing the font.

---

## Round 0.2 · accepted · Accept

I am pleased to confirm that your paper has been accepted for publication in PeerJ.

Thank you for submitting your work to this journal.

·

Basic reporting

No comment

Experimental design

No comment

Validity of the findings

No comment

Additional comments

The authors have addressed all suggested revisions which I believe improved the quality of this publication.
The results presented in this manuscript are novel and useful for the fish farming industry and other researchers working on fish microbiome. As a result, I believe that this article should be accepted.

Reviewer 2 ·

Basic reporting

The authors have properly revised the manuscript based on all comments.

Experimental design

The authors have properly revised the manuscript based on all comments.

Validity of the findings

The authors have properly revised the manuscript based on all comments.

Additional comments

The authors have properly revised the manuscript based on all comments.